# A Semantic Topology Graph to Detect Re-Localization and Loop Closure of the Visual Simultaneous Localization and Mapping System in a Dynamic Environment

**DOI:** 10.3390/s23208445

**Published:** 2023-10-13

**Authors:** Yang Wang, Yi Zhang, Lihe Hu, Wei Wang, Gengyu Ge, Shuyi Tan

**Affiliations:** 1School of Computer Science and Technology, Chongqing University of Posts and Telecommunications, Chongqing 400065, China; d200201021@stu.cqupt.edu.cn (Y.W.); d200201006@stu.cqupt.edu.cn (L.H.); d190201021@stu.cqupt.edu.cn (W.W.); d190201004@stu.cqupt.edu.cn (G.G.); d220201037@stu.cqupt.edu.cn (S.T.); 2Advanced Manufacturing and Automatization Engineering Laboratory, Chongqing University of Posts and Telecommunications, Chongqing 400065, China

**Keywords:** semantic topology graph, Visual SLAM, ORB-SLAM2, dynamic environment, mobile robots

## Abstract

Simultaneous localization and mapping (SLAM) plays a crucial role in the field of intelligent mobile robots. However, the traditional Visual SLAM (VSLAM) framework is based on strong assumptions about static environments, which are not applicable to dynamic real-world environments. The correctness of re-localization and recall of loop closure detection are both lower when the mobile robot loses frames in a dynamic environment. Thus, in this paper, the re-localization and loop closure detection method with a semantic topology graph based on ORB-SLAM2 is proposed. First, we use YOLOv5 for object detection and label the recognized dynamic and static objects. Secondly, the topology graph is constructed using the position information of static objects in space. Then, we propose a weight expression for the topology graph to calculate the similarity of topology in different keyframes. Finally, the re-localization and loop closure detection are determined based on the value of topology similarity. Experiments on public datasets show that the semantic topology graph is effective in improving the correct rate of re-localization and the accuracy of loop closure detection in a dynamic environment.

## 1. Introduction

Simultaneous localization and mapping (SLAM) is one of the core problems in mobile robotics research [1,2]. Compared to laser sensors, vision sensors have the advantages of fine perception, low price, smaller size, and lighter weight. Thus, Visual SLAM (VSLAM) has made great progress in the last few decades. Among various VSLAM algorithms, feature-based algorithms are widely used in long-term robot deployment because of their high efficiency and scalability. However, most existing SLAM systems rely on hand-crafted visual features, such as SIFT [3], the Shi–Tomasi method [4], and ORB [5], which may not provide consistent feature detection and association results in dynamic environments. For example, when either the scene or the viewpoint has been changed, ORB-SLAM2 frequently fails to recognize previously visited scenes because of less visual feature information [6]. Thus, the mobile robot needs to use re-localization and loop closure detection to identify whether this scene has been visited before. Generally, the re-localization is accomplished using loop closure detection to correlate information. Firstly, the bag-of-words model is used to extract candidate frames for relocation with high similarity to the current frame. Then, the frame-to-frame matching method is used to match the local features of the current frame and re-localize candidate frames until all the candidate frames are traversed or the information is associated. However, in dynamic environments, Visual SLAM often fails in re-localization and loop closure detection due to the insufficient numbers of matched local feature point pairs. However, the numbers of matched local feature point pairs are insufficient because of dynamic objects, which can vastly impair the performance of the Visual SLAM system [7]. Therefore, Visual SLAM often fails in re-localization and loop closure detection. In order to address this challenging topic, some researchers have performed some work on feature removal [8,9]. The stability of the system’s visual odometer is enhanced through removing feature points from dynamic objects.

In the real dynamic scene, re-localization and loop closure detection will fail with limited feature information [10]. Therefore, the mobile robot often needs to go back to the previous place to extract more feature information, so as to complete the feature information matching to realize re-localization and loop closure detection. The combination of Visual SLAM and deep learning can solve this problem better than traditional methods [11]. The main idea of deep learning is to extract image features using a network trained in advance [12]. However, the disadvantage of deep learning is that it is computationally intensive and requires high-performance equipment. Furthermore, it is difficult to construct suitable models to store high-dimensional feature vectors. Thus, in recent years, researchers have attempted to introduce semantic information into Visual SLAM [13,14]. Using semantic information to describe the environment can effectively simplify the process of saving and comparing environmental information [15,16].

ORB-SLAM2 is the most widely used of the Visual SLAM frameworks [6]. The re-localization and loop closure detection of the ORB-SLAM2 framework are mainly accomplished by using the current frame to match feature points with the candidate keyframes. Therefore, the numbers of extracted feature points determine the accuracy and matching speed of re-localization and loop closure detection. The more feature points are extracted, the more accurate the localization and the faster the corresponding speed. However, it will affect the feature point extraction if the dynamic object moves fast in a dynamic environment, which will affect the accuracy of re-localization and loop closure detection. Despite this, the relative positions and distances of static objects do not change, irrespective of the motion of dynamic objects in the dynamic environment. Thus, we can build a semantic topology graph using the relative positions of static objects to assist in re-localization and loop closure detection, instead of relying only on static feature points.

Therefore, based on the above problems, this paper proposes a Visual SLAM method with semantic topology based on ORB-SLAM2. The proposed method improves the failure of re-localization and loop closure detection in dynamic environments due to limited feature information. The specific improvements are as follows:(1)The method of object detection is used to obtain information about static objects in the dynamic environment.(2)Low feature information and frame loss may occur due to the occlusion of dynamic objects. Thus, this paper proposes to construct a topological graph by utilizing the property of the invariant spatial position of static objects by judging the similarity of the semantic topological graph to find similar keyframes and then quickly determining its own pose information.(a)Semantic nodes are obtained from the central points of static objects.(b)The edges between nodes are obtained using the Delaunay triangulation method.(c)An innovative topological graph similarity comparison algorithm is achieved.

This paper is organized as follows: Section 2 presents the related work, and Section 3 is a description of the method, which first summarizes the system structure of this paper and then describes the algorithm of this paper in detail. Section 4 is the experimental section, which first verifies the feasibility of the method by using different data and then tests and evaluates the re-localization and loop closure detection. Section 5 is the conclusion.

## 2. Related Work

### 2.1. Re-Localization

ORB-SLAM2, as a sparse feature-based VSLAM method, is prone to tracking loss in position estimation. The re-localization function of ORB-SLAM2 is activated when frames are lost. The re-localization is realized by discriminant coordinate regression. This method uses the PnP algorithm to obtain the essential matrix after using RANSAC to obtain the fundamental matrix describing the relationship between two image positions, and then uses the PnP algorithm to obtain the essential matrix jointly with the camera internal parameters to solve the camera poses [17]. The purpose is to obtain a sufficient number of matched feature points from the before and after sequence frame images for rotation and translation to solve the camera poses so that the lost camera poses can be estimated and the tracking process can be resumed [18]. The core idea of re-localization is to find the keyframe that is closest to the current frame among the previous keyframes. Firstly, the BoW of the current frame must be calculated. Then, the frames with high similarity and a higher number of matching feature points than 15 in the BoW model are determined as the sequence of candidate keyframes. Finally, the success of re-localization is determined by whether the number of interior points matching the current frame with the candidate frames is greater than 50. The specific process is shown in Figure 1. Due to the presence of dynamic objects in the environment, incorrect matching information often occurs during feature matching.

### 2.2. Loop Closure Detection (LCD)

Loop closure detection (LCD) is the ability of a mobile robot to recognize a scene that has been reached and to realize loop closure. The basic process is to calculate the similarity by comparing between keyframes and determine whether they pass through the same place or “return to the origin”. The LCD problem is the process of determining the correlation between current and historical data to identify whether a location has been reached before. The essence of the LCD problem is to reduce the cumulative error in map construction [19]. With the development of computer vision, the LCD algorithm based on appearance information has become the mainstream algorithm in the early stage, among which BoVW is the most common algorithm [20]. BoVW has been widely used in the LCD of VSLAM systems because of its high detection efficiency and retrieval accuracy. However, the presence of dynamic objects will interfere with the judgment of LCD [21]. In recent years, the continuous development of deep learning technology in the fields of image recognition, computer vision, and mobile robotics has provided new solution ideas for the LCD module in SLAM systems. DS-SLAM [22] and SegNet [23] were employed to segment dynamic objects. DS-SLAM removed the feature points located in the area of the dynamic scene to alleviate dynamic interference in loop detection. The authors of reference [24] proposed to realize LCD by integrating visual–spatial–semantic information with features of a topological graph and a convolutional neural network. The authors first built semantic topological graphs based on semantic information and then used random walk descriptors to characterize the topological graphs for graph matching. Finally, the authors calculated the geometric similarity and the appearance similarity to determine the loop closure detection. Reference [25] presents a strategy that models the visual scene as a semantic sub-graph by only preserving the semantic and geometric information from object detection. The authors used a sparse Kuhn–Munkres algorithm to speed up the search for correspondence among nodes. The shape similarity and the Euclidean distance between objects in the 3D space were leveraged and united to measure the image similarity through graph matching. However, topology building is more complex, and these studies have only involved LCD without considering the re-localization of a mobile robot. Inspired by the above modules, the re-localization and loop closure detection methods based on a semantic topology graph are proposed in this paper.

## 3. Methodology

### 3.1. Method Overview

To solve the problem of re-localization and loop closure detection failing due to the interference of dynamic objects, we constructed a topology graph for assisted localization based on ORB-SLAM2 using the static regions obtained from object detection. Five threads run in parallel in the proposed system: tracking, semantic topology graph, local map, loop closing, and full BA. The framework of the system is shown in Figure 2. The raw RGB images are processed in the tracking thread and semantic topology graph thread simultaneously. The tracking thread first extracts ORB feature points and waits for the image that has a classification of the dynamic properties via object detection. Then, ORB feature point outliers are judged and removed based on the classification of the dynamic properties. The ORB feature points in the highly dynamic region and low dynamic region are considered outliers. The semantic topology graph thread is mainly designed to build a semantic topology graph using the results of object detection. The detailed process of the semantic topology graph is shown in Figure 3. The RGB images first are classified for dynamic properties by object detection. This process is described in Section 3.2. Then, we establish the semantic topology graph with a low dynamic static region. This process is described in Section 3.3. In order to reduce the computation load, we created a semantic topology graph only for the keyframe. Furthermore, the semantic topology graph was saved as a bag of topology graphs, which is convenient for loop-closing threads and re-localization threads.

This paper proposes to use the nature of the invariant spatial position of static objects to construct the semantic topography graph. By judging the similarity of semantic topology, we can find similar key frames and then quickly determine their own pose information. The process of the proposed solution is presented in Figure 3. Firstly, we extracted semantic information using the YOLOv5 network. The low dynamic regions based on the classification of the dynamic properties are saved as static regions. Secondly, the topology graph was obtained using Delaunay triangulation. Furthermore, we acquired the information of semantic nodes and edges in the topology separately. Finally, we obtained the weight of topologies with the information of semantic nodes and edges, which is used to compute similarity.

### 3.2. Object Detection

The robustness of SLAM is improved so that dynamic objects are identified and eliminated by the semantic information. For instance, among the same type of SLAM systems, DynaSLAM [26], Detect-SLAM [27], and DS-SLAM [22] introduce instance segmentation, object detection, and semantic segmentation, respectively. Each of these methods has its own advantages. They are capable of greatly improving the performance of SLAM by detecting and removing dynamic objects. Each of these methods has its own advantages. Table 1 shows the brief relationship between segmentation accuracy and efficiency of different methods. As part of the vSLAM optimization process, dynamic objects are usually detected and then treated as outliers. However, semantic segmentation networks are computationally expensive, making them impractical for real-time or robotic applications. Thus, object detection methods are widely used in the preprocessing stage. In reference [8], for instance, YOLOv4 is adopted to predict classes and bounding boxes of objects in real time. Then, the dynamic object probability model is added to enhance the real-time performance of the ORB-SLAM2 system. Similarly, Theodorou, C. et al. [9] proposed a VSLAM system based on ORB-SLAM3 and on YOLOR. This system uses the object detection models YOLOX and YOLOR to detect moving objects and extract feature points. On the basis of the results and semantic data in the image, a module is introduced that can remove dynamic objects. Lastly, the results show that the accuracy of this system is significantly improved in dynamic indoor environments.

In this paper, we are more concerned with the segmentation efficiency of neural networks, so it was decided to introduce the YOLOv5 network. YOLOv5 is a robust and efficient model that can detect target objects well in images [28]. Compared with YOLOv3 and YOLOv4, this model has higher accuracy and better real-time performance. The main structure of the YOLOv5 network consists of a feature extractor, a multi-scale feature fusion module, a prediction head, and a post-processing module. The feature extractor uses the CSPNet structure, which can quickly extract image features [29]. The module of multi-scale feature fusion can improve the detection accuracy through the fusion of feature maps with different resolutions. The prediction head consists of a classifier and a repressor, which are used to quickly predict the location and category of the detection box. The post-processing module can remove redundant detection results by performing non-extreme value suppression operations on the data.

We obtained the training model trained on YOLOv5 through the MS COCO datasets [30]. The training data of the MS COCO datasets include more than 80 different classes of objects, which is basically sufficient for the use of VSLAM in dynamic environments. The result of the object detection using YOLOv5 is shown in Figure 4. As can be seen from the object detection results, the various categories of objects, such as people, chairs, monitors, keyboards, mice, and cups, were successfully detected using YOLOv5. Based on life experience, we carried out a simple classification of motion for various different categories of objects. It is shown in Table 2. Furthermore, there is an overlapping part between the person and the chair in the picture, and the YOLOv5 algorithm can successfully detect it. At the same time, some relatively small objects, such as keyboards, mice, and books, also were successfully detected. Therefore, the overall detection results are in line with the application requirements of VSLAM.

### 3.3. Establishment of Topology Graph

Firstly, we obtained the object detection frame by detecting the original frame using YOLOv5 and retaining the static objects. Secondly, we computed the central points of static objects boxes, which are used as the vertices of Delaunay triangulation. Finally, the topology graph was obtained according to Delaunay triangulation. The process is shown in Figure 5.

#### 3.3.1. Establishment of Semantic Nodes

The result of the image object detection contains the object class, information about the detection frame coordinates, and confidence values. The center point of the object detection box is used as the vertex of the topology grape. Thus, for each detected object in an image, we represent it as a triple: (1)$$o_{i}=\left(m_{i},z_{i},c_{i}\right),$$
where $m_{i}=\left(u_{i},v_{i}\right)$, $u_{i}$, $v_{i}$ represents the horizontal and vertical coordinates of the center point of the enclosing object $o_{i}$ box. $z_{i}$ is the depth information at the $\left(u_{i},v_{i}\right)$ position in the depth image that corresponds to the current image. $c_{i}$ is the class label of the object. In accordance with the bounding box, the geometric center $r_{i}=\left(u_{i},v_{i}\right)$ of the detected object is defined:(2)$$\left\{\begin{array}{c} u_{i}=\frac{x_{i1}-x_{i2}}{2}+x_{i1}\\ v_{i}=\frac{y_{i1}-y_{i2}}{2}+y_{i1} \end{array}\right.,$$
where $\left(x_{i1},y_{i1}\right)$ and $\left(x_{i2},y_{i2}\right)$ are the coordinates of the upper left and lower right corner of the bounding box, which are relative to the pixel coordinate system. In order to facilitate matrix similarity calculation, numerical numbers are used to represent the categories of objects identified after object detection, and their corresponding relationships are shown in the following Table 3. Therefore, $c_{i}$ takes the value of the class label number corresponding to the different classes in Table 3.

The set of vertex information in each image is represented as:(3)$$O={\left[o_{i}\right]}_{i=1,2\ldots ,n},$$
where *n* represents the numbers of categories in the image.

#### 3.3.2. Establishment of Edges

We obtained the topology graph using Delaunay triangulation from all nodes in the image [31]. The generated graph structure is similar to a sparse mesh, as shown in Figure 6a. In Delaunay triangulation, for a given set of discrete points, triangulation is carried out such that no point is inside the circumcircle of any triangle. It maximizes the minimum angle of all of the angles of the triangles. Therefore, only adjacent feature points are connected in the graph. Furthermore, Delaunay triangulation is applied to reduce the complexity of the construction of the topology graph [32]. On the other hand, the topology graph using Delaunay triangulation does not change if the relative positions of the vertices do not change. This is one of the reasons why the Delaunay triangulation method is used to construct the topology in the method of this paper: the positions of static objects do not change in the dynamic environment, so the relative positions of the static objects detected by object detection do not change either. In other words, the static topology obtained by Delaunay triangulation is unique if the position of the static object remains unchanged in a image. At the same time, Delaunay triangulation is regional; that is, adding, deleting, or moving a vertex only affects the neighboring triangles. As shown in Figure 6, Figure 6a has one more target point *P* than Figure 6b. Furthermore, the line of other points does not change except for the line of point *P* and its neighboring points. For instance, Figure 6b shows that the yellow line is newly increased and the red line remains unchanged. This means that the topology graph between the other points is not changed. Therefore, the similarity of the frames can be judged by verifying the similarity of the topology graph in the next step, on account of the unique and regional topology graph.

Because the relative distance between static objects in the scene does not change with the positions of the bounding boxes, in order to better calculate the similarity of topology structure, we calculated the distance value of the line between vertices. The relative distance $d_{ij}$ between $o_{i}$ and $o_{j}$ in three-dimensional space was calculated using Euclidean distance according to the following formula:(4)$$d_{ij}=\left\{\begin{array}{cc} {∥o_{i}-o_{j}∥}_{2}, & o_{i}\;\mathrm{is}\;\mathrm{connected}\;\mathrm{to}\;o_{j}\\ 0, & o_{i}\;\mathrm{is}\;\mathrm{not}\;\mathrm{connected}\;\mathrm{to}\;o_{j} \end{array}\right.,$$
where ${∥o_{i}-o_{j}∥}_{2}=\sqrt{{\left(X_{i}-X_{j}\right)}^{2}+{\left(Y_{i}-Y_{j}\right)}^{2}+{\left(Z_{i}-Z_{j}\right)}^{2}}$. $\left(X_{i},Y_{i},Z_{i}\right)$ is the camera coordinate of point $o_{i}$.

The camera coordinate of point $o_{i}$ is as follows:(5)$$\left\{\begin{array}{c} Z_{i}=\frac{z_{i}}{F}\\ X_{i}=\frac{\left(u_{i}-c_{x}\right)Z_{i}}{f_{x}}\\ Y_{i}=\frac{\left(v_{i}-c_{y}\right)Z_{i}}{f_{y}} \end{array}\right.,$$
where value $z_{i}$ is the depth value at position of point $o_{i}\left(u_{i},v_{i}\right)$ in the aligned depth image. F is the scaling factor for the depth image. Furthermore, $f_{x}$,$f_{y}$,$c_{x}$,$c_{y}$ are the intrinsic parameters of the camera.

Thus, in the current graph, the correlation matrix *D* is used to describe the relationships of edges in the topology.
(6)$$D={\left[d_{i}\right]}_{i=1,2,\cdots ,n},d_{i}=\left(d_{i1},d_{i2},\cdots ,d_{in}\right),$$
where *n* indicates the numbers of categories in the image.

#### 3.3.3. The Topology Graph Representation

The topology graph H=(O,D) is established for each frame, where *O* and *D* represent the set of vertices’ information and the correlation matrix of the edges in the topology graph, respectively. In particular, the vertices of our topology graph are built on the basis of the results of object detection. Every vertex has its own attribute, as described in Equation (Equation 3), and the numbers of vertices are equal to the numbers of the bounding boxes. All vertices are connected to each other using undirected edges. The edges of a topological graph represent the distance between two objects.

Thus, based on Equations (Equation 3) and (Equation 6), we can obtain $H_{p}=\left(O_{p},D_{p}\right)$ for frame $I_{p}$, where $O_{p}$ is an n×4 matrix, and *n* represents the numbers of categories obtained after object detection in frame $I_{p}$. $O_{p}$ is defined as:(7)$$\begin{array}{c} O_{p}=\left[\begin{array}{c} o_{1}\\ o_{2}\\ \vdots \\ o_{n} \end{array}\right]=\left[\begin{array}{cccc} u_{1} & v_{1} & z_{1} & c_{1}\\ u_{2} & v_{2} & z_{2} & c_{2}\\ \vdots & \vdots & \vdots & \vdots \\ u_{n} & v_{n} & z_{n} & c_{n} \end{array}\right] \end{array}$$

$D_{p}$ is an n×n correlation matrix with weights, and *n* represents the numbers of categories obtained after object detection in frame $I_{p}$. $D_{p}$ is defined as based on Equation (Equation 6):(8)$$\begin{array}{c} D_{p}=\left[\begin{array}{cccc} 1 & d_{12} & \cdots & d_{1n}\\ d_{21} & 1 & \cdots & d_{2n}\\ \vdots & \vdots & \ddots & \vdots \\ d_{n1} & d_{n2} & \cdots & 1 \end{array}\right] \end{array}$$

### 3.4. Similarity Comparison Algorithm

In order to calculate the similarity of topologies in the two images more efficiently, we define $W_{p}$ to denote the weights of topologies in frame $I_{p}$. $W_{p}$ is defined as:(9)$$\begin{array}{cc} W_{p} & =D_{p}O_{p}\\ & =\left[\begin{array}{cccc} 1 & d_{12} & \cdots & d_{1n}\\ d_{21} & 1 & \cdots & d_{2n}\\ \vdots & \vdots & \ddots & \vdots \\ d_{n1} & d_{n2} & \cdots & 1 \end{array}\right]\cdot \left[\begin{array}{cccc} u_{1} & v_{1} & z_{1} & c_{1}\\ u_{2} & v_{2} & z_{2} & c_{2}\\ \vdots & \vdots & \vdots & \vdots \\ u_{n} & v_{n} & z_{n} & c_{n} \end{array}\right]\\ \; & =\left[\begin{array}{cccc} \omega_{11} & \omega_{12} & \omega_{13} & \omega_{14}\\ \omega_{21} & \omega_{22} & \omega_{23} & \omega_{24}\\ \vdots & \vdots & \vdots & \vdots \\ \omega_{n1} & \omega_{n2} & \omega_{n3} & \omega_{n4} \end{array}\right]\\ & ={\left[\begin{array}{c} \omega_{i} \end{array}\right]}_{i=1,2,\cdots ,n} \end{array},$$
where $\omega_{i}=\left(\omega_{i1},\omega_{i2},\omega_{i3},\omega_{i4}\right)$, and *n* is the number of nodes in frame $I_{P}$. Therefore, the $W_{P}$ is an n×4 matrix. Similarly, the weight $W_{Q}$ of the topology graph in frame $I_{Q}$ is:(10)$$\begin{array}{cc} W_{Q} & =D_{Q}O_{Q}={\left[\begin{array}{c} \omega_{i} \end{array}\right]}_{i=1,2,\ldots ,k} \end{array},$$
where *k* denotes the number of nodes in frame $I_{Q}$.

This paper uses the cosine similarity to express the relative differences between the topology of the two frames $I_{P}$ and $I_{Q}$. Since the number of nodes in each frame is uncertain, the numbers of nodes in frames $I_{P}$ and $I_{Q}$ are not necessarily equal. Thus, before calculating the cosine similarity, we need to ensure that $W_{p}$ and $W_{Q}$ have the same dimension. The dimension is labeled *N*, which is defined as:(11)$$N=\left\{\begin{array}{cc} n, & n\geq k\\ k, & n<k \end{array}\right.$$

We first need to calculate the cosine angle of $W_{p}$ and $W_{Q}$ with the same dimension. Then, we take the weighted average sum of the cosine angle matrix. We consider the value of weighted average $S(I_{P},I_{Q})$ as the similarity of matrix $W_{p}$ and $W_{Q}$. $S(I_{P},I_{Q})$ is defined as:(12)$$S\left(I_{P},I_{Q}\right)=\frac{1}{N}\sum_{}\frac{W_{p}\cdot W_{Q}}{\left|\right|W_{p}\left|\right|\times \left|\right|W_{Q}\left|\right|}$$

In summary, a semantic topology graph of static objects is established using the above method. Ultimately, the similarity of the two images is judged by calculating the value of $S(I_{P},I_{Q})$.

## 4. Experiment

### 4.1. Datasets

In this study, the effect of the proposed algorithm was evaluated using the Technical University of Munich (TUM) RGB-D datasets and OpenLORIS-Scene datasets, which are shown in Table 4. The TUM datasets are the most used SLAM data in the literature. There are both high- and low-dynamic office sequences in the TUM RGB-D datasets [33,34] recorded with a Microsoft Kinect sensor at full frame rate (30 Hz). RGB (640 × 480), depth images, and ground-truth trajectory were recorded with a high-accuracy motion capture system.

The OpenLORIS-Scene datasets by Xuesong Shi et al. are designed to be tested for the real-world practicality of lifelong SLAM algorithms for service robots [35]. OpenLORIS-Scene is a recently published dataset. It provides real-world robotic data with more challenging factors and significant environmental changes such as blurred, featureless images and dim lighting. Environmental changes are likely to be the main challenge to re-localization. We mainly used the Office series of OpenLORIS-Scene to evaluate the algorithm, specifically seven sequences including dynamic objects (persons) in a university office with benches and cubicles. The images of the TUM RGBD datasets and OpenLORIS-Scene datasets are shown in Figure 7.

### 4.2. Calculated Similarity

We selected four keyframes from “freiburg3_w_xyz” to verify the feasibility of the proposed topology in this paper, as shown in Figure 8. Figure 8a is the first keyframe, and the topology graph is constructed based on the object detection results. Similarly, Figure 8b–d are the topology graphs of the 22nd, 44th, and 197th keyframes, respectively.

In order to verify the correctness of the proposed method, we used the 22nd, 44th, and 197th keyframes to calculate the similarity with the first keyframe. At the same time, we also carried out the similarity calculation with the first keyframe and itself. The obtained results are shown in Table 5.

### 4.3. Re-Localization Evaluation

Re-localization is often formed as a pipeline of image retrieval followed by relative pose estimation, similar to LCD but often with a much larger database of candidate images and with more emphasis on high recall as opposed to high precision of LCD. In this subsection, we refer to the experiments of Reference [35].


*Correctness Score of Re-Localization (CS-R)*


In order to better evaluate the performance of re-localization, the authors proposed a score to evaluate the correctness of re-localization [35]. The score is called the correctness score of re-localization (CS-R), which is defined as follows:(13)$$C^{\varepsilon ,\phi }S^{\tau }-R=e^{-\frac{t_{0}-t_{min}}{\tau }}\cdot C^{\varepsilon ,\phi }\left(P_{0}\right),$$
where τ is a scaling factor, and Reference [35] suggests to set τ=60 s. The absolute trajectory error (ATE) threshold ε and absolute orientation error (AOE) threshold ϕ should be set according to the area of the scene and the expected drift of the SLAM algorithm. Furthermore, $\left(t_{0}-t_{min}\right)$ is the algorithm initialization.

$C^{\varepsilon ,\phi }\left(P_{0}\right)$ is the correctness of the position at time $t_{0}$. For each pose, $p_{k}$ is estimated at time $t_{k}$, given the ground-truth pose at that time, and we assess the correctness of the estimate according to its ATE and AOE.
(14)$$C^{\varepsilon ,\phi }\left(P_{\kappa }\right)=\left\{\begin{array}{cc} 1, & if\;ATE\left(P_{\kappa }\right)\leq \varepsilon \;and\;AOE\left(P_{\kappa }\right)\leq \phi \\ 0, & otherwise \end{array}\right.$$

Reference [35] applied Equation (Equation 13) to verify the correctness of the re-localization on datasets office2 and office7 of OpenLORIS-Scene. The authors of Reference [35] suggest setting ε=0.3 and ϕ=∞. Thus, we performed a similar comparison test based on Reference [35]. The results were compared with the data provided in the reference, as follows in Table 6.

From the results of the correct rate, our method is better than the traditional method. However, the proposed method in this paper is dependent on the results of object detection. Our experimental results are slightly lower than those of DXSLAM [36], a global feature-based image retrieval and group matching method.


*Re-localization test*


To test whether algorithms could continuously achieve re-localization in changed scenes, we entered the seven data of the office in turn. We first counted whether the different methods could accomplish re-localization on seven sets of data. The results are shown in Table 7. “•”and “∘” indicate successful and unsuccessful re-localization, respectively. Experimental results show that most algorithms fail to re-localize on the fifth sequence. Our analyses consider that the light is too dark, which affects the feature points and object extraction. Thus, re-localization fails eventually.

Then, we documented whether algorithms could consistently perform the correct pose estimation. The results are shown in Figure 9. The office sequence has seven sets of data, and each black dot on the top line represents the start of one data sequence in Figure 9. For the four different algorithms in Figure 9, the blue dots indicate successful re-localization, while red crosses are unsuccessful re-localization. The blue line and red line indicate correct and incorrect pose estimation, respectively. The experimental results show that re-localization fails on the second, fifth, and seventh sequences for most algorithms. Similarly, it was because of the light problem that affected the results of the experiment.

Finally, we evaluate the average accuracy on the seven sets of data of the office sequence. The results are shown in Table 8. A larger average accuracy means a more robust approach. The statistical results show that the average accuracy of the method in this paper is 60.8%, which is slightly higher than the other methods, so that indicates the good robustness of our proposed method.

### 4.4. Loop Closure Detection Evaluation

In this study, in order to verify the effectiveness of the proposed method, the VSLAM system was improved based on the ORB-SLAM2 algorithm and utilized in the mobile robot platform of the built VSLAM system. As shown in Figure 10, the software control of the mobile robot is realized by the ROS kinetic software system (ubuntu18.04). The vision sensor equipped in this mobile robot is an Intel RealSense D435I depth camera. The camera is capable of acquiring RGB and depth information with high resolution and low latency. The camera employs the latest depth perception technology, which not only enables high-precision depth measurement in both indoor and outdoor environments but also supports application scenarios such as dynamic object tracking and gesture recognition. Its pixel resolution and depth perception distance are 1280 × 720 and 0.105–10 m, respectively.

We conducted experiments with DS-SLAM, DynaSLAM, ORB-SLAM2, and our methods and plotted the precision–recall curve. The results are shown in Figure 11. It can be seen from the figure that our experimental results are slightly better than those of the other methods. High accuracy is also guaranteed in the case of high recall. In order to evaluate the accuracy of the algorithms on loop closure detection, we calculated the accuracy of loop closure detection about the different methods on the mobile robot experimental platform. We judge the number of correct loops in the detection results by manual labeling. We then express the accuracy by dividing the number of correct loopbacks by the total number of identifications. The results are shown in Table 9. From the statistical results, it can be seen that the traditional methods BOW and ORB-SLAM2 are less low due to the interference of dynamic objects, while our method is slightly better than the DS-SLAM and DynaSLAM algorithms for dynamic scenes. It can be seen that the algorithm in this paper improves the accuracy of the loop closure detection.

At the same time, we created the trajectory plots of the mobile robot platform. Figure 12a shows the keyframe trajectory without the closed loop. Figure 12b shows a loop closure detection result obtained by the proposed method, which is shown by the red line. It can be seen that the proposed method could still detect a large number of loop closures under the influence of dynamic scenes. Finally, we obtained the keyframe trajectory after applying the proposed method, which is shown in Figure 12c. In order to evaluate the accuracy of the algorithms on loop closure detection, we calculated the accuracy of loop closure detection about the different methods on the mobile robot experimental platform. We judge the number of correct loops in the detection results from manual labeling. We then express the accuracy by dividing the number of correct loopbacks by the total number of identifications. The results are shown in Table 9. From the statistical results, it can be seen that the traditional methods BOW and ORB-SLAM2 are less low due to the interference of dynamic objects, while our method is slightly better than DS-SLAM and DynaSLAM algorithms for dynamic scenes. It can be seen that the algorithm in this paper improves the accuracy of the loop closure detection.

## 5. Conclusions

Re-localization and loop closure detection in dynamic environments, due to limited feature points information, often fail, and the correct re-localization and recall of loop closure detection are also low when the mobile robot loses frames in a dynamic environment. So, this paper proposes the re-localization and loop closure detection method with a semantic topology graph based on ORB-SLAM2. We conducted experiments on public datasets such as TUM, OpenLORIS-Scene, and a self-made platform. The results clarify that our method can improve the feasibility, accuracy, and stability of the VSLAM system in dynamic scenes. However, the proposed method relies heavily on the results of object detection. Whether the detected objects are sufficiently reliable greatly impacts our experimental performance. At the same time, the detector model may hardly predict correct results when there are significant differences between training scenes and actual scenes. In future work, we can employ self-supervised or unsupervised deep learning approaches in order to overcome this issue. On the other hand, the “bag of topology graphs” will take up storage space, so we will also further improve the real-time performance of the system.

## Figures and Tables

**Figure 1 sensors-23-08445-f001:**
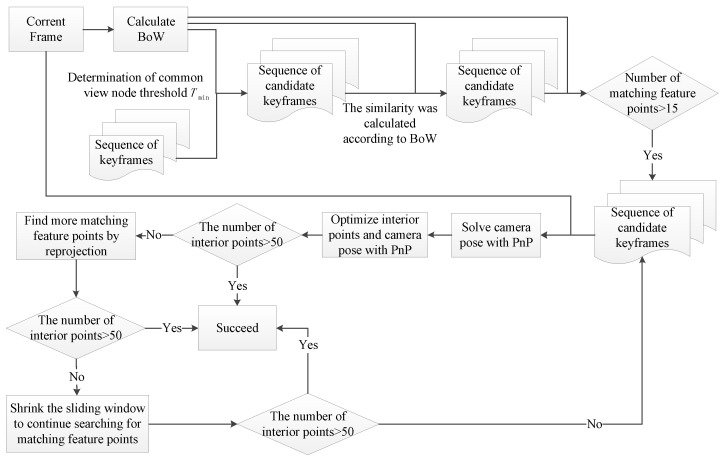
The process of re-localization.

**Figure 2 sensors-23-08445-f002:**
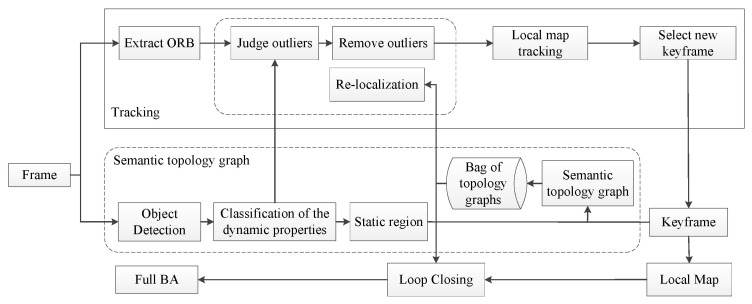
The framework of the proposed Visual SLAM system.

**Figure 3 sensors-23-08445-f003:**
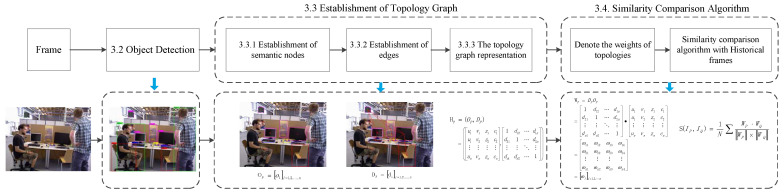
The process of the proposed solution.

**Figure 4 sensors-23-08445-f004:**
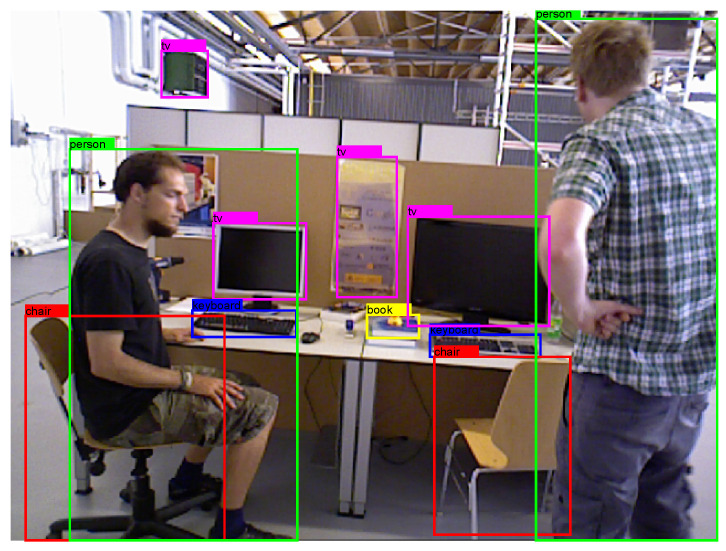
The result of the object detection using YOLOv5.

**Figure 5 sensors-23-08445-f005:**
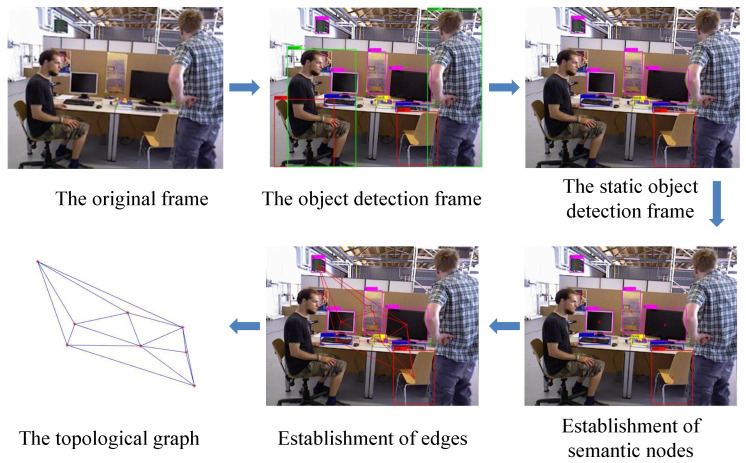
The obtaining process of topological graph.

**Figure 6 sensors-23-08445-f006:**
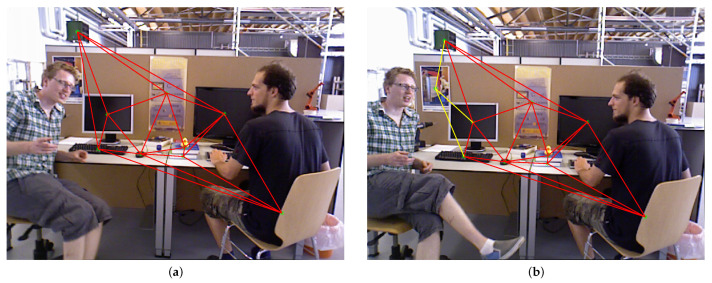
The process of creating topological edges.

**Figure 7 sensors-23-08445-f007:**
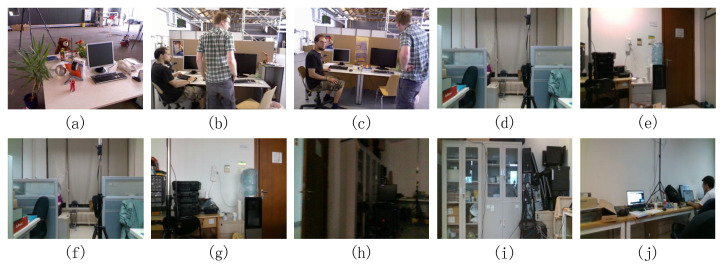
The images of the TUM RGB-D and OpenLORIS-Scene. (**a**–**c**), respectively, are “freibureg2_desk_with_person”, “freiburg3_walking_rpy”, and “freiburg3_walking_xyz” in TUM RGB-D; (**d**–**j**) are in turn the pictures of the seven sequences in OpenLORIS-Scene.

**Figure 8 sensors-23-08445-f008:**
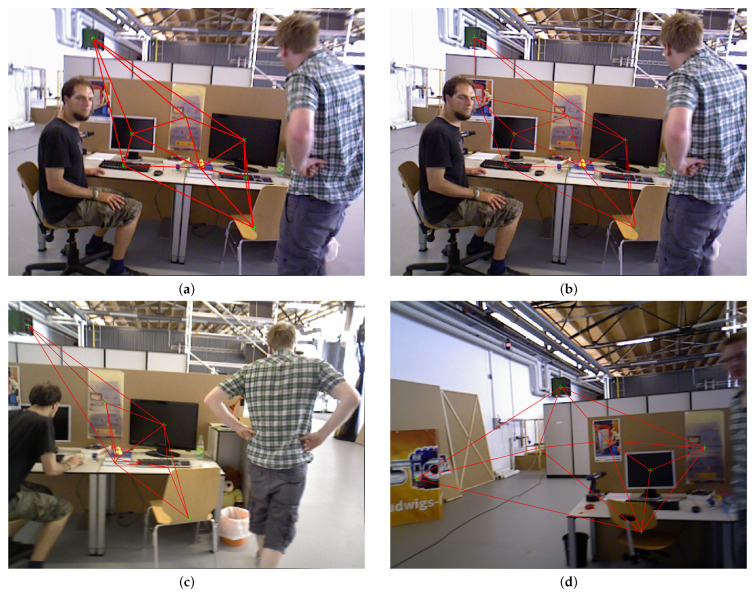
The topology graph of different frames.

**Figure 9 sensors-23-08445-f009:**
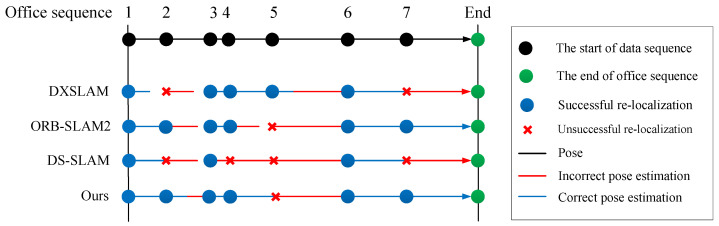
Whether the algorithms consistently performed correct pose estimation.

**Figure 10 sensors-23-08445-f010:**
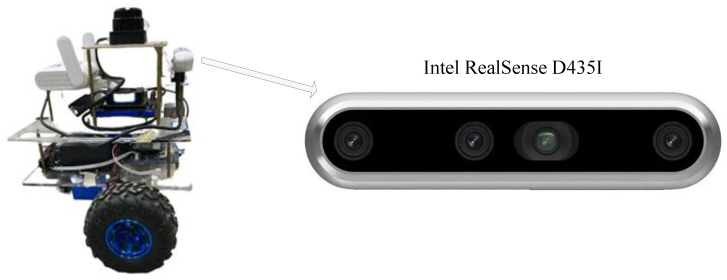
Service robot platform physical picture and Intel RealSense D435I depth camera.

**Figure 11 sensors-23-08445-f011:**
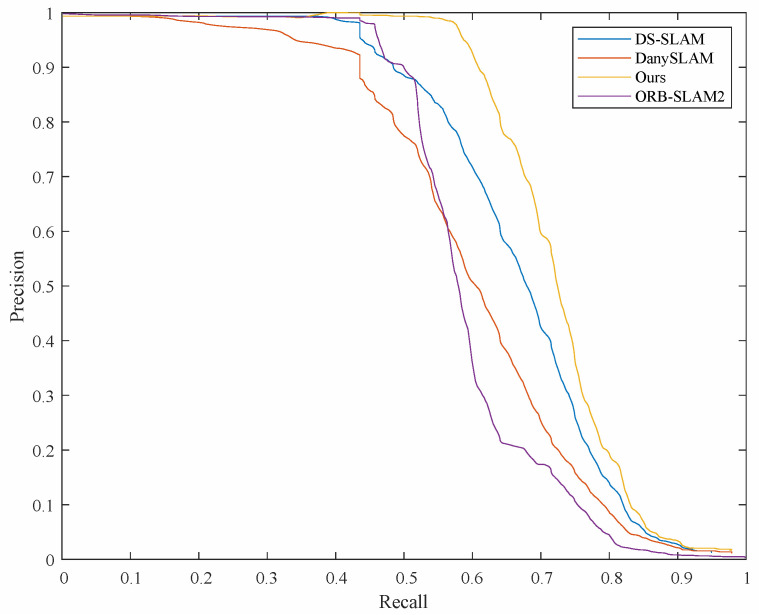
Precision–recall curve.

**Figure 12 sensors-23-08445-f012:**
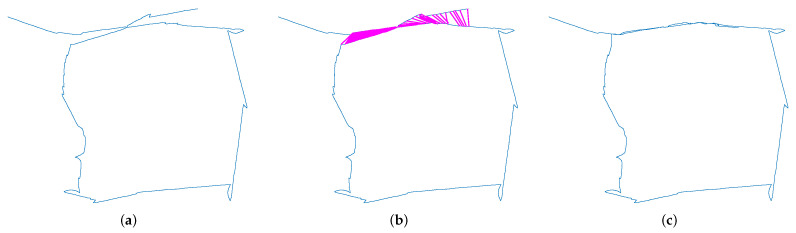
The keyframe trajectory with loop closure detection for the city center. (**a**) The keyframe trajectory without loop closure detection. (**b**) The results of loop closure detection. The red lines indicate the detected closed loop. (**c**) The keyframe trajectory with loop closure detection using the method of this paper.

**Table 1 sensors-23-08445-t001:** Brief relationship between segmentation accuracy and efficiency of different methods.

Method	Segmentation Accuracy	Segmentation Efficiency
object detection	low	high
semantic segmentation	middle	middle
instance segmentation	high	low

**Table 2 sensors-23-08445-t002:** Classification of the dynamic properties of common objects in life.

Classification	Objects
Highly Dynamic	People
Medium Dynamic	Chairs, Books
Low Dynamic	Desks, TVs

**Table 3 sensors-23-08445-t003:** The value of the $c_{i}$ class label.

Class	Class Label Number	Applicable Dataset
tv	1	fre2, fre3, office
keyboard	2	fre2, fre3, office
chair	3	fre2, fre3, office
book	4	fre2, fre3, office
mouse	5	fre2
teddy_bear	6	fre2
potted_plant	7	fre2
cup	8	fre2, office
vase	9	fre2
car	10	fre2, office
desk	11	office
water_dispenser	12	office
bucket	13	office
door	14	office
bag	15	office
bookshelf	16	office

“fre2” represents “freiburg2_desk_with_person”; “fre3” represents “freiburg3_walking_rpy” and ”freiburg3_walking_xyz”; “office” represents OpenLORIS-Scene.

**Table 4 sensors-23-08445-t004:** The datasets used in this paper.

	Datasets	Camera	Images
TUM	freiburg2_desk_with_person	RGB-D	4067
	freiburg3_walking_rpy		910
	freiburg3_walking_xy		859
OpenLORIS-Scene (Office)	Office1	RGB-D	809
	Office2		899
	Office3		360
	Office4		870
	Office5		1589
	Office6		1080
	Office7		1141

**Table 5 sensors-23-08445-t005:** The similarity values between different keyframes and the first keyframe.

Similarity	1th	22th	44th	197th
1th	100%	95%	55%	68%

**Table 6 sensors-23-08445-t006:** The value of CS-R for different methods.

Office2,7	ORB-SLAM2	DS-SLAM	DXSLAM	Dynamic-SLAM	Ours
CS-R	0.997	0.996	0.999	0.996	0.998

**Table 7 sensors-23-08445-t007:** Whether algorithms could re-localize.

Office Sequence	Sequence 1	Sequence 2	Sequence 3	Sequence 4	Sequence 5	Sequence 6	Sequence 7
DXSLAM	•	∘	•	•	•	•	∘
ORB-SLAM2	•	•	•	•	∘	•	•
DS-SLAM	•	∘	•	∘	∘	•	∘
Ours	•	•	•	•	∘	•	•

“•” is successful re-localization; “∘” is unsuccessful re-localization.

**Table 8 sensors-23-08445-t008:** The average accuracy on office sequence for different methods.

Office Sequence	ORB-SLAM2	DS-SLAM	DXSLAM	Ours
Average accuracy	52.5%	28.6%	54.6%	60.8%

**Table 9 sensors-23-08445-t009:** The accuracy for different methods on the mobile robot experimental platform.

	BoW	ORB-SLAM2	DS-SLAM	DynaSLAM	Ours
Accuracy	61.8%	69.3%	76.2%	78.4%	80.1%

## Data Availability

Data available in a publicly accessible repository that does not issue DOIs Publicly available datasets were analyzed in this study. These data can be found here: https://cvg.cit.tum.de/data/datasets/rgbd-dataset/download (accessed on 1 October 2023); https://shimo.im/docs/HhJj6XHYhdRQ6jjk/read (accessed on 1 October 2023).

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
