# Peer review of "A Semantic Topology Graph to Detect Re-Localization and Loop Closure of the Visual Simultaneous Localization and Mapping System in a Dynamic Environment"

_sensors, 2023, doi:10.3390/s23208445_

Round 1
Reviewer 1 Report
The paper proposes a semantic topology graph to improve re-localisation in dynamic environments. The evaluation based on publicly available dataset compares the performance with prior work. The authors could justify the choice of ORB-Slam2, and whether the approach is applicable to more recent algorithms. The results show that the proposed approach is only marginally better, so authors need to emphasize better the significance of contribution. Fig 9 results seem to be based just on one scenario. The robot based evaluation can be a strong point but this section requires much more details. When discussing the dynamic SLAM challenges, the authors could provide more context, including the work on dynamic object removal [1][2]. The conclusion section needs improvement, e.g. by adding details on further work. Finally, the paper is readable but needs more clarity and proofreading for grammar issues.
[1] Ai YB, Rui T, Yang XQ, He JL, Fu L, Li JB, Lu M. Visual SLAM in dynamic environments based on object detection. Defence Technology. 2021 Oct 1;17(5):1712-21.
[2] C. Theodorou, V. Velisavljevic, V. Dyo, Visual SLAM for Dynamic Environments Based on Object Detection and Optical Flow for Dynamic Object Removal, Sensors 2022, 22(19), 7553
Please see detailed comments.
Author Response
1. “The results show that the proposed approach is only marginally better, so authors need to emphasize better the significance of contribution.”
In order to prove the accuracy of the proposed method, we enrich the experimental content.
2. “Fig 9 results seem to be based just on one scenario. The robot based evaluation can be a strong point but this section requires much more details.”
The data in Figure 9 comes from office's sequence data, which consists of a total of seven sets of data. Because there is a connection between the sequences, we draw the seven results from each method in a straight line. Therefore, each row in Figure 9 represents the experimental results of one method on seven data sets.
To better explain the experimental results of the different methods on the office dataset, we added the experiments in Table 7 and Table 8, respectively. The advantages of proposed method are explained in terms of repositioning and accuracy.
3. “When discussing the dynamic SLAM challenges, the authors could provide more context, including the work on dynamic object removal [1][2].”
Thank you for your comments, these two documents is very much in line with my research. Therefore, I have included them as reference 7,8 and added them to Part 1 and Section 3.2
4. “The conclusion section needs improvement, e.g. by adding details on further work.”
We have added some specific work in the conclusion section. In future work, we can employ self-supervised or unsupervised deep learning approaches in order to overcome this issue. On the other hand, "Bag of topology graphs" will take up storage space, so we will also further improve the real-time performance of the system.
5. “the paper is readable but needs more clarity and proofreading for grammar issues.”
We have edited the full text in English.
Reviewer 2 Report
Authors are presenting a SLAM system based on proposed semantic topology graph approach.
After introduction that included relevant references and motivation for this research, methodolog of the proposed approach is described. Unfortunately, although this should be the most important part of the paper, it is non informative enough. Clarity of presentation is low. Figures 1 and 2 should be described and explained in details. Connections and data flows between modules in Fig 2 should be revisited and explained. Steps of the procedure presented in Figure 3 should be connected to the subsections in the following text. There are lot of sentences that were interrupted and separated into 2 sentences without any explainable reason (for example in lines 129, 130, 137, 140, 207). Therefore, text is very hard to follow. Some typos also occuras well as unusual literature referencing.
Ending of the section 3 is sudden, it seems that some conluding sentences are missing.
Results section should also be significantly improved and extended. Some subsection titles should be changed (4.3.2) and some results should be corrected. For example similarity score of the frame with the same frame is calculated as 1% in table 5.
Language needs thorough proofreading.
Author Response
We have re-presented the system framework diagram (Figure 2). And we also have added descriptions of the contents of Figures 1 and 2 in Section 2.1 and 3.1. At the same time, we have modified Figure 3. And each link has been corresponding to a subsection in the text. Because of carelessness issues, the article does have a problem with literature citation errors. We have checked and revised the references and grammar issues in the full text.
The section 3 focuses on describing the specific computational procedures of the methodology. So, we've added a small summary statement at the end.
We have expanded the experiments in subsections 4.3 and 4.4. A description of re-localization is added to demonstrate the stability of the method. In order to prove that the accuracy rate of the method is higher, we also counted the accuracy rate of different methods on the office data set and the experimental platform. The experimental results are shown in the table 8 and table 9. Finally, in the loopback detection section, we showed the P-R curves to surface the advantages of the method proposed in the article. At the same time, because the content of subsections 4.3.2 is to explain the feasibility of the proposed method, the experimental content is less and the title is not reasonable. So, we have merged the contents of subsections 4.3.1 and 4.3.2, which makes subsection 4.3 more substantial. Finally, in order to better understand the results of the experiment, we have expressed the data in table 5 as a percentage and modified it.
Round 2
Reviewer 1 Report
The revised version has improved significantly. I am satisfied with the corrections.
The paper is clearly written in most parts. However, it would still benefit from proofreading for grammar mistakes to check for minor errors such as in line 419.
Author Response
Thank you very much for your comments and suggestions. We have made some modifications to the English language.
Reviewer 2 Report
Authors have improved the first version of the paper significanly. I have no further comments. Just a note, written answer of the authors was too brief and not informative enough. After just reading it, one could think that only a small changes have been done. But, in fact, quite significant changes has been made.
English has been significantly improved. Some minor style issues were found, nothing critical.
Author Response
Thank you very much for your comments and suggestions. Next, we supplemented the written answer for the first version of the paper. Firstly,we have strengthened the description of Figure 1, which is the process of re-localization. The added content is on lines 103 to 109 of the text. Then, In Section 3.1, we re-present the system framework diagram. And we revisited and explained the connections and data flows between modules in Figure 2. And we add describe the contents and processes in the system framework in detail. Finally, we have modified Figure 3. And each link has been corresponding to a subsection in the text.
Some revisions have also been made to the English language.